# High sensitivity of ultrasound for the diagnosis of tuberculosis in adults in South Africa: A proof-of-concept study

**Matthew Fentress**[1]*, **Patricia C. Henwood**[2], **Priya Maharaj**[3], **Mohammed Mitha**[3], **Dilshaad Khan**[3], **Philip Caligiuri**[4,5], **Aaron S. Karat**[1], **Stephen Olivier**[6], **Anita Edwards**[6], **Dirhona Ramjit**[6], **Nokwanda Ngcobo**[6], **Emily B. Wong**[6,7], **Alison D. Grant**[1,8,9]

**1** TB Centre, London School of Hygiene & Tropical Medicine, London, United Kingdom, **2** Thomas Jefferson University, Philadelphia, Pennsylvania, United States of America, **3** Department of Pulmonology and Critical Care, Inkosi Albert Luthuli Central Hospital, Durban, South Africa, **4** Department of Radiology and Imaging Sciences, University of Utah, Salt Lake City, Utah, United States of America, **5** Veterans Affairs Salt Lake City Health Care System, Salt Lake City, Utah, United States of America, **6** Africa Health Research Institute, KwaZulu-Natal, South Africa, **7** Division of Infectious Diseases, Heersink School of Medicine, University of Alabama at Birmingham, Birmingham, Alabama, United States of America, **8** Africa Health Research Institute, School of Laboratory Medicine and Medical Sciences, College of Health Sciences, University of KwaZulu-Natal, KwaZulu-Natal, South Africa, **9** School of Public Health, University of the Witwatersrand, Johannesburg, South Africa

* Mfentress2010@gmail.com

**Data Availability Statement:** The primary data that support the findings of this study are available via

## Abstract

### Background

There are limited data on the performance characteristics of ultrasound for the diagnosis of pulmonary tuberculosis in both HIV-positive and HIV-negative persons. The objective of this proof-of-concept study was to determine the sensitivity and specificity of ultrasound for the diagnosis of tuberculosis in adults.

### Methods

Comprehensive thoracic and focused abdominal ultrasound examinations were performed by trained radiologists and pulmonologists on adults recruited from a community multimorbidity survey and a primary healthcare clinic in KwaZulu-Natal Province, South Africa. Sputum samples were systematically collected from all participants. Sensitivity and specificity of ultrasound to detect tuberculosis were calculated compared to a reference standard of i) bacteriologically-confirmed tuberculosis, and ii) either bacteriologically-confirmed or radiologic tuberculosis.

### Results

Among 92 patients (53 [58%] male, mean age 41.9 [standard deviation 13.7] years, 49 [53%] HIV positive), 34 (37%) had bacteriologically-confirmed tuberculosis, 8 (9%) had radiologic tuberculosis with negative bacteriologic studies, and 50 (54%) had no evidence of active tuberculosis. Ultrasound abnormalities on either thoracic or abdominal exams were detected in 31 (91%) participants with bacteriologic tuberculosis and 27 (54%) of those

an online repository available at Africa Health Research: https://doi.org/10.23664/AHRI.Pocus.AnalyticalDataset.2022.v1.

**Funding:** This work was supported, in whole or in part, by the Bill & Melinda Gates Foundation (Grant Number OPP1212544). AG and EW are the PI and co-PI, respectively, for this grant. The funders were involved in the design of the study, but had no role in data collection, data analysis, decision to publish or preparation of the manuscript. Under the grant conditions of the Foundation, a Creative Commons Attribution 4.0 Generic License has already been assigned to the Author Accepted Manuscript version that might arise from this submission.

**Competing interests:** The authors have no competing interests to declare.

without tuberculosis. Sensitivity and specificity of any ultrasound abnormality for bacteriologically-confirmed tuberculosis were 91% (95% confidence interval [CI] 76%–98%) and 46% (95% CI 32%–61%). Sensitivity and specificity of any ultrasound abnormality for either bacteriologically-confirmed or radiologic tuberculosis were 86% (95% CI 71%–95%) and 46% (95% CI 32%–61%). Overall performance did not appear to differ markedly between participants with and without HIV.

## Conclusion

A comprehensive ultrasound scanning protocol in adults in a high TB burden setting had high sensitivity but low specificity to identify bacteriologically-confirmed tuberculosis.

## Introduction

Tuberculosis (TB) is one of the leading causes of death from a single infectious agent worldwide and was responsible for 1.5 million deaths in 2020 [1]. Priorities to reduce TB incidence and mortality include new diagnostic tests that are rapid, affordable, and easy to use at the point of care [2]. Current standard diagnostics for TB include sputum microscopy, sputum nucleic acid amplification tests (NAATs), and chest radiography. World Health Organization (WHO) guidance supports chest radiography as a component of systematic screening for active TB [3]. However, microscopy has poor sensitivity [4], and NAAT and chest radiography are generally not available at primary healthcare level, where TB screening and triage is most often undertaken in TB-prevalent settings.

Thoracic ultrasound can detect abnormalities of the pleura and parenchyma and can been used for diagnosis of pneumonia with excellent performance characteristics compared to conventional radiography [5–7]. Recent advances in ultrasound technology have produced affordable, portable machines that can be transported with relative ease to rural or resource-constrained settings. Clinicians can be trained in brief courses to acquire and interpret pulmonary images [8], and ultrasound examinations can be performed rapidly at the point of care. These features make thoracic ultrasound a potentially promising novel diagnostic tool for TB, particularly as an alternate to chest radiography in regions where radiography is not available.

A 2021 systematic review of thoracic ultrasound for TB diagnosis found only six, mostly hospital-based studies with methodological limitations, including small sample sizes and a lack of well characterized comparison groups [9]. Several small studies have demonstrated that pulmonary lesions can be detected by thoracic ultrasound in patients with TB [9–14], including consolidations, cavitary lesions, and a miliary pattern. Two of these studies reported thoracic ultrasound findings in a high proportion (96%–100%) of patients with pulmonary TB (PTB) [11, 12], while another found that several thoracic ultrasound features were associated with diagnosis of PTB [13]. In addition, several findings on abdominal and cardiac ultrasound have high specificity for extrapulmonary TB in HIV-positive individuals in hospital settings [15–17]. If ultrasound has appropriate performance characteristics for diagnosis of PTB, it could be incorporated into point-of-care algorithms for triage of TB, and potentially shorten the time to treatment initiation. However, the current data on the performance characteristics of ultrasound for the diagnosis of TB remain limited.

The aim of this proof-of-concept study was to determine the performance characteristics of thoracic and abdominal ultrasound for the diagnosis of TB in adults compared to a microbiological reference standard, with a view to determining whether larger scale evaluation is

justified. To better represent people requiring TB services in clinics or communities, the study focused on ambulatory rather than hospitalized patients.

## Methods

### Study design and participant selection

We conducted a cross-sectional study in KwaZulu-Natal, South Africa from October 2019 to February 2020. Adults aged 18 years and older were recruited from two sources: i) a community multimorbidity survey [18, 19] that involved systematic investigations for TB and ii) a primary healthcare clinic where TB is diagnosed and treated. *Community survey*: As part of a separate study, all adult residents in a demographic surveillance area in rural KwaZulu-Natal were invited to take part in a multimorbidity survey [18]. This is an area with an estimated HIV prevalence of 30% among adults [20] and TB case notification rate of 394 per 100,000 in 2018 [21]. Those who participated underwent a digital chest radiograph which was analyzed using version 5 of the Computer Assisted Diagnosis for TB (CAD4TB) (Diagnostic Image Analysis Group, The Netherlands) and subsequently read by an experienced radiologist. Survey participants who reported any TB symptom (cough, weight loss, night sweats, or fever) or had a CAD4TB score above 25 were asked to give a sputum specimen, which was divided into two in the laboratory; one portion was tested using NAAT (Xpert MTB/RIF Ultra, Cepheid, Sunnyvale, CA, USA), and the other was cultured on liquid media (MGIT, Becton Dickinson Microbiology Systems, Cockeysville, MD). We invited a systematic sample of community survey participants to enroll in this ultrasound study if, at the time of the community survey, they had a) a sputum sample positive for *Mycobacterium tuberculosis (Mtb)* on Xpert MTB/RIF Ultra or culture, or b) they had a chest radiograph with a CAD4TB score over 65 and defined as abnormal by the radiologist AND a sputum sample negative for *Mtb*; or c) they had a normal chest radiograph as defined by the radiologist, a negative sputum sample, and reported neither TB symptoms nor a prior history of TB. *TB clinic*: A convenience sample of adults at a primary healthcare clinic near Durban starting TB treatment within the previous 7–14 days who had a positive sputum NAAT result were invited to take part in this ultrasound study.

Exclusion criteria for all participants included known or suspected drug-resistant TB, as defined by rifampin resistance on Xpert MTB/RIF results; too unwell to undergo study procedures; and known or suspected pregnancy. The decision to exclude people with drug-resistant TB was based on operational considerations. Enrolled participants had prolonged close contact with study staff shortly after initiation of treatment. Since there is lower certainty that individuals with drug-resistant TB will rapidly become noninfectious after treatment initiation compared to drug-sensitive TB, participants with drug-resistant TB were excluded to reduce the risk of TB transmission.

### Study procedures

Participants drawn from the community survey gave permission for demographic and clinical data, sputum microbiology and chest radiographs from the survey to be used in this study. Participants from the TB clinic underwent sputum Xpert MTB/RIF testing during their routine evaluation and, as part of the study, 1) completed a questionnaire concerning health and care history, including TB symptoms and HIV and TB treatment, similar to the questions asked in the community survey, and 2) underwent chest radiography, since this is not performed routinely for people with TB in South Africa. All participants gave venous blood for testing for HIV antibodies. Each participant underwent comprehensive thoracic and focused abdominal ultrasounds performed according to the study protocol by clinicians masked to all clinical and

imaging data (described below). Each chest radiograph was reviewed by an experienced International Labour Organization (ILO) certified reader masked to all clinical and imaging data.

## Case definitions

Bacteriologic TB was defined as either NAAT (Xpert MTB/RIF Ultra) or culture positive for *Mtb* from a sputum sample. Radiologic TB was defined as typical chest radiography features of pulmonary or extrapulmonary TB as determined by the study radiologist. The primary definition of an abnormal ultrasound, chosen to maximize sensitivity, was the presence of any one or more of the following: consolidation, small subpleural consolidation (SPC), cavity, B-lines, irregular pleural line, pleural effusion, pericardial effusion, splenic lesions, abdominal lymphadenopathy, hepatic lesions, or ascites (Table 1). Additional ultrasound composites, hypothesized *a priori* to have improved specificity over the primary definition of an abnormal ultrasound, are defined in Table 1.

## Chest radiography

Probable or definite active radiological TB was defined as a spectrum of confidence on the part of the reader that active TB was present. This was based on the identification of abnormalities including cavities, infiltrates, or consolidation typically in the upper and/or middle zones of the lung, a miliary pattern, and/or mediastinal or hilar adenopathy. If radiological signs of previous TB were present, lower certainty was assigned to the classification of active TB.

**Table 1. Classification of ultrasound findings.**

| THORACIC FINDINGS | Ultrasound Finding | Definition |
|---|---|---|
| | Small subpleural consolidation (SPC) | Hypoechoic subpleural region less than 10 mm x 10 mm, with distinct borders and trailing comet-tail artefacts |
| | Consolidation | Subpleural, echo-poor, or tissue-like region >10 mm in depth or length, with or without sonographic air bronchograms |
| | Cavitation | Consolidation >10mm in depth or length with hypoechoic central clearing. If color flow is used, no color flow present within the hypoechoic central clearing. |
| | Pleural effusion | Anechoic collection between the pleural line or diaphragm and the chest wall. |
| | B-lines | B-1 pattern: 3 or more B-lines in 1 intercostal space. |
| | | B-2 pattern: Diffuse or confluent B-lines in 1 region. |
| | Irregular pleural line | Pleural line abnormal contour or thickened when imaging performed with probe at 90-degree angle to pleura. |
| ABDOMINAL FINDINGS | Pericardial effusion | Echo-free space between hyperechoic pericardium and myocardium. |
| | Ascites | Extraluminal echo-free space in peritoneum |
| | Periaortic lymphadenopathy | Hypoechoic round or ovoid structure in periaortic region >1.0 cm confirmed to not be vascular structure by dynamic imaging techniques |
| | Splenic lesions | Hypoechoic or hyperechoic round or ovoid lesions within the spleen |
| | Hepatic lesions | Hypoechoic or hyperechoic round or ovoid lesions within the liver |
| COMPOSITE SCORES | Any Thoracic Pathology | One or more of consolidation, SPC, cavity, any B1 or B2 pattern, irregular pleural line, or pleural effusion |
| | Thoracic Combo 1 | One or more of consolidation, SPC, cavity, diffuse B1, or B2 pattern with subpleural granularity or pleural effusion |
| | Thoracic Combo 2 | One or more of upper region*consolidation, SPC, cavity, diffuse B1, or B2 pattern with subpleural granularity |
| | Thoracic Combo 3 | One or more of consolidation, ≥2 SPC, cavity, diffuse B1, or B2 pattern with subpleural granularity |
| | Any FASH Pathology | One or more of pericardial effusion, splenic lesions, periaortic lymphadenopathy, hepatic lesions, ascites |
| | FASH Combo 1 | One or more of pericardial effusion, splenic lesions, periaortic lymphadenopathy. |

* Upper region defined as upper 1/3 of chest wall anteriorly or posteriorly

† FASH = Focused Assessment with Sonography for HIV-Associated Tuberculosis

## Ultrasound methodology

The thoracic scanning technique included a systematic interrogation of each intercostal space according to previously described techniques [11–13], in both transverse and longitudinal planes, from lung apices to diaphragm (GE Logiq P9, Boston, MA, USA). The chest was divided into nine regions on each side–upper, middle, and lower for each of the anterior, lateral, and posterior aspects of the chest–for a total of 18 regions per participant. Representative clips were saved from each region, and each region was classified according to the presence or absence of the following findings: sub-pleural consolidation (SPC), consolidation, cavity, pleural effusion, B-lines, and irregular pleural line (Table 1).

Abdominal ultrasound scanning was performed according to previously published techniques for Focused Assessment with Sonography for HIV-Associated Tuberculosis (FASH) [22]. Standardized views were obtained, representative clips were saved from each view, and the exam was coded by the presence or absence of the following findings: pericardial effusion, ascites, periaortic lymphadenopathy, hepatic lesions, and splenic lesions (Table 1).

Thoracic ultrasounds were performed by a board-certified radiologist with substantial prior experience of performing ultrasound examination, but not previously trained in thoracic ultrasound, and three pulmonologists, each of whom participated in a thoracic ultrasound training program tailored to the study protocol. Abdominal ultrasound examinations were performed exclusively by the board-certified radiologist who was trained to scan according to the study standard operating procedure. All anterior and lateral ultrasound scanning was performed in the supine position. Posterior scanning was performed in the sitting position or, if necessary for patient comfort, in the lateral decubitus position. Thoracic ultrasound scanning was performed in the "Abdominal" setting, with tissue harmonics turned off, the focal point adjusted to the region of the pleural line, and manual time-gain compensation buttons set to the midline. Depth was initially set to 11 cm and, when necessary, was adjusted to optimize image acquisition of pathologic findings.

All saved ultrasound images were reviewed by at least one ultrasound expert (MF, PH) masked to patient diagnosis and clinical data. If the masked reviewer agreed with the initial read, the initial read was used as the final interpretation for that region. If the masked reviewer disagreed with the initial read, the region in which there was disagreement was reviewed by a second masked expert. If both masked reviewers agreed (i.e., if both disagreed with the initial read), their read was used as the final interpretation. If the second masked reviewer agreed with the initial read, the initial read was used as the final interpretation for that region. In cases where the second masked reviewer disagreed with both the initial read and the first masked reviewer, the final interpretation for that region was reached by consensus agreement between the two masked reviewers.

## Data analysis

For the primary analysis, the sensitivity and specificity of ultrasound to detect TB was calculated a) compared to a reference standard of bacteriologically-confirmed TB: in this analysis, participants with chest radiographs typical of active TB but negative on bacteriologic testing were excluded; and b) compared to a reference standard of either bacteriologically-confirmed TB or radiologic TB. For secondary analyses, we calculated the sensitivity and specificity for combinations of ultrasound abnormalities that were hypothesized to have improved specificity over the primary definition, defined *a priori* by researchers with substantial prior experience of ultrasound for TB (MF, PH), and for individual thoracic and abdominal ultrasound findings (Table 1). Descriptive statistics were used to characterize demographic and clinical data. In a *post hoc* analysis we compared findings in participants with confirmed TB to a subset of participants without

TB restricted to those who had no evidence of current or previous TB, i.e., with a normal chest radiograph, negative sputum culture, and no symptoms or past history of TB. Sensitivity of chest radiograph to identify TB was calculated compared to a reference standard of bacteriologically-confirmed TB. We did not estimate specificity of chest radiography because individuals were selected for our "healthy" control group based on having a normal chest radiograph.

Sample size was based on precision estimates. With 50 participants with bacteriologically-confirmed TB, we calculated that we would be able to demonstrate ultrasound sensitivity of 80% with a 95% confidence interval (CI) of 67%–89%, and with 100 participants without active TB, we would be able to demonstrate specificity of ultrasound of 80% with a 95% CI of 71%–87%. The study was designed to be exploratory, with a relatively small number of participants, aiming to estimate sensitivity and specificity relatively imprecisely to guide whether larger-scale evaluation was warranted. We did not calculate predictive values because our sample purposively included more people with active TB than would usually be found in routine populations being screened for TB, and thus predictive values from this study could not be generalized. Data analysis was performed in R (Version 4.1.0) with the 'epiR' package used to estimate exact 95% binomial CI for sensitivity and specificity.

## Ethical considerations

Ethical approval for this study and the community survey was obtained from the Biomedical Research Ethics Committee, University of KwaZulu-Natal, South Africa, and from the London School of Hygiene & Tropical Medicine, London, United Kingdom. Written or witnessed verbal informed consent was obtained from all participants.

## Results

### Participant characteristics

Overall, 189 eligible participants were screened: due to coronavirus disease 2019 (COVID-19) restrictions, enrollment was terminated early, and 92 participants were enrolled in the study. A total of 34 participants had bacteriologic TB, eight had radiologic TB, and 50 had no evidence of active TB (Table 2). Of those enrolled, 61 participants were from the community survey (three bacteriologic TB, eight radiologic TB, 50 no TB), and 31 participants were from the primary care clinic (31 bacteriologic TB). The mean age of the 92 participants was 41.9 (standard deviation 13.7) years, 53 (58%) were male, 20 (22%) had a history of prior TB treatment, and 49 (53%) were HIV positive (Table 2).

### Ultrasound findings

Ultrasound abnormalities on either thoracic or abdominal exams were detected in 31 (91%) participants with bacteriologic TB and 27 (54%) of those without TB. Thoracic ultrasound was abnormal in 29 (85%) of participants with bacteriologic TB and 24 (48%) of those without TB. Sonographic consolidation and SPC were detected in 22 (65%) and 19 (56%) of participants with TB, and in 5 (10%) and 14 (24%) of those without TB (Table 3 and Fig 1). Median (interquartile range) scan time for thoracic ultrasound was 32 (27–40) minutes, and for abdomen was 7 (6–9.5) minutes.

### Diagnostic accuracy of ultrasound

In the primary analysis with the reference standard of bacteriologic TB, sensitivity and specificity of any thoracic or FASH ultrasound abnormality for TB was 91% (95% CI 76%–98%) and 46% (95% CI 32%–61%) (Table 3). Thoracic ultrasound alone demonstrated a sensitivity and

**Table 2. Demographic and clinical characteristics of study population (N = 92).**

| Characteristic | Total (N = 92) | Active Tuberculosis (N = 42) | No Active Tuberculosis | |
| --- | --- | --- | --- | --- |
| | | | Abnormal CXR (N = 13) | Healthy (N = 37) |
| Male, n (%) | 53 (58) | 27 (64) | 7 (54) | 19 (51) |
| Age (years), mean (SD) | 41.9 (13.7) | 39 (11.2) | 49.2 (9.6) | 42.2 (16.7) |
| Diabetic, n (%)* | 3 (3.3) | 1(2.4) | 0 (0) | 2 (5.4) |
| Smoker, n (%) | 21 (23) | 14 (33) | 4 (31) | 3 (8.1) |
| Prior tuberculosis treatment, n (%) | 20 (22) | 7 (17) | 13 (100) | 0 (0) |
| Body Mass Index (kg/m$^2$), mean (SD) | 23.2 (6.7) | 20.6 (4.71) | 22.5 (4.4) | 26.3 (8.1) |
| Symptoms of tuberculosis†, n (%) | 34 (37) | 31 (74) | 3 (23) | 0 (0) |
| HIV positive, n (%) | 49 (53) | 27 (64) | 9 (69) | 13 (35) |
| CD4 count (cells/μL), median (IQR) | 541 (409, 818) | 541(470, 744) | 920 (405, 1124) | 468 (354, 684) |
| On ART‡, n (%) | 35/48 (73) | 14/27 (51.8) | 9/9 (100) | 12/13 (92) |

* Self-reported.

† One or more of the following: cough, weight loss, night sweats, fever

‡ Self-reported; 48 of 49 participants had answered this question.

ART: antiretroviral therapy; CXR: chest radiograph; HIV: human immunodeficiency virus; IQR: interquartile range; SD: standard deviation

specificity of 85% (95% CI 69%–95%) and 52% (95% CI 37%–66%), while any abdominal ultrasound finding (FASH) had a sensitivity and specificity of 24% (95% CI 11%–41%) and 88% (95% CI 76%–95%). Additional combinations of thoracic ultrasound findings showed a sensitivity ranging from 68%–76% and a specificity ranging from 72%–84% (Table 3). Amongst HIV-positive participants, sensitivity and specificity of any thoracic or FASH ultrasound abnormality for TB was 90% (95% CI 68%–99%) and 32% (95% CI 14%–55%), while amongst HIV-negative participants it was 93% (95% CI 66%–100%) and 57% (37%–76%) (S1 Table).

When either bacteriologic or radiographic TB was used as the reference standard, sensitivity and specificity of any ultrasound abnormality for TB were 86% (95% CI 71%–95%) and 46% (95% CI 32%–61%) (Table 3). Thoracic ultrasound alone demonstrated a sensitivity of 81% (95% CI 66%–91%) and specificity of 52% (95% CI 37%–66%), while any abdominal ultrasound finding (FASH) had a sensitivity and specificity of 24% (95% CI 12%–39%) and 88% (95% CI 76%–95%).

In a *post hoc* analysis comparing participants with bacteriologic TB to only those participants who were healthy with normal chest radiographs, sensitivity and specificity of any ultrasound abnormality for TB were 91% (95% CI 76%–98%) and 54% (95% CI 37%–71%) (S2 Table).

Compared to the reference standard of bacteriologic TB, the sensitivity and specificity of sonographic lung consolidation were 65% (95% CI 46%–80%) and 90% (95% CI 78%–97%), of SPC was 56% (95% CI 38%–73%) and 72% (95% CI 58%–84%), and of cavity was 9% (95% CI 2%–24%) and 98% (95% CI 89%–100%) (Table 3). Upper lung region consolidation had sensitivity and specificity of 38% (95% CI 22%–56%) and 100% (95% CI 93%–100%), while upper lung region SPC had sensitivity and specificity of 26% (95% CI 13%–44%) and 88% (95% CI 76%–95%). Pericardial effusion had sensitivity and specificity of 18% (95% CI 7%–35%) and 88% (95% CI 76%–95%). Performance characteristics for other individual findings are reported in Table 3.

### Chest radiograph

Amongst the 34 participants with bacteriologic TB, any abnormality on chest radiograph had a sensitivity of 85% (95% CI 69%–95%), and any abnormality classified as definite or probable TB had a sensitivity of 56% (95% CI 38%–73%).

**Table 3. Ultrasound findings in all participants stratified by TB status.**

| Ultrasound finding no. (%) | Bacteriologic TB | | | | Bacteriologic or Radiologic TB | | | |
|---|---|---|---|---|---|---|---|---|
| | TB | No TB | *Sensitivity (95% CI*)* | *Specificity (95% CI)* | TB | No TB | *Sensitivity (95% CI)* | *Specificity (95% CI)* |
| | *(N = 34)* | *(N = 50)* | | | *(N = 42)* | *(N = 50)* | | |
| **A. Composite Scores** | | | | | | | | |
| Any Thoracic or FASH[†] Pathology[^] | 31 (91) | 27 (54) | 0.91 (0.76–0.98) | 0.46 (0.32–0.61) | 36 (86) | 27 (54) | 0.86 (0.71–0.95) | 0.46 (0.32–0.61) |
| Any Thoracic Pathology[^] | 29 (85) | 24 (48) | 0.85 (0.69–0.95) | 0.52 (0.37–0.66) | 34 (81) | 24 (48) | 0.81 (0.66–0.91) | 0.52 (0.37–0.66) |
| Thoracic Combo 1[^] | 26 (76) | 14 (28) | 0.76 (0.59–0.89) | 0.72 (0.58–0.84) | 31 (74) | 14 (28) | 0.74 (0.58–0.86) | 0.72 (0.58–0.84) |
| Thoracic Combo 2[^] | 23 (68) | 14 (28) | 0.68 (0.49–0.83) | 0.72 (0.58–0.84) | 28 (67) | 14 (28) | 0.67 (0.5–0.8) | 0.72 (0.58–0.84) |
| Thoracic Combo 3[^] | 25 (74) | 8 (16) | 0.74 (0.56–0.87) | 0.84 (0.71–0.93) | 30 (71) | 8 (16) | 0.71 (0.55–0.84) | 0.84 (0.71–0.93) |
| FASH Combo 1[^] | 7 (21) | 6 (12) | 0.21 (0.09–0.38) | 0.88 (0.76–0.95) | 9 (21) | 6 (12) | 0.21 (0.1–0.37) | 0.88 (0.76–0.95) |
| Thoracic Combo 1 or FASH Combo 1[^] | 27 (79) | 19 (38) | 0.79 (0.62–0.91) | 0.62 (0.47–0.75) | 32 (76) | 19 (38) | 0.76 (0.61–0.88) | 0.62 (0.47–0.75) |
| Any FASH Pathology[^] | 8 (24) | 6 (12) | 0.24 (0.11–0.41) | 0.88 (0.76–0.95) | 10 (24) | 6 (12) | 0.24 (0.12–0.39) | 0.88 (0.76–0.95) |
| **B. Specific Findings** | | | | | | | | |
| Consolidation | 22 (65) | 5 (10) | 0.65 (0.46–0.8) | 0.9 (0.78–0.97) | 27 (64) | 5 (10) | 0.64 (0.48–0.78) | 0.9 (0.78–0.97) |
| Consolidation, Upper | 13 (38) | 0 (0) | 0.38 (0.22–0.56) | 1 (0.93–1) | 16 (38) | 0 (0) | 0.38 (0.24–0.54) | 1 (0.93–1) |
| Small Subpleural Consolidation (SPC) | 19 (56) | 14 (24) | 0.56 (0.38–0.73) | 0.72 (0.58–0.84) | 24 (57) | 14 (24) | 0.57 (0.41–0.72) | 0.72 (0.58–0.84) |
| SPC, Upper | 9 (26) | 6 (12) | 0.26 (0.13–0.44) | 0.88 (0.76–0.95) | 12 (29) | 6 (12) | 0.29 (0.16–0.45) | 0.88 (0.76–0.95) |
| SPC, 2 or more | 13 (38) | 7 (14) | 0.38 (0.22–0.56) | 0.86 (0.73–0.94) | 18 (43) | 7 (14) | 0.43 (0.28–0.59) | 0.86 (0.73–0.94) |
| B1 or B2 pattern | 5 (15) | 9 (18) | 0.15 (0.05–0.31) | 0.82 (0.69–0.91) | 7 (17) | 9 (18) | 0.17 (0.07–0.31) | 0.82 (0.69–0.91) |
| B2 pattern | 0 (0) | 2 (4) | 0 (0–0.1) | 0.96 (0.86–1) | 0 (0) | 2 (4) | 0 (0–0.08) | 0.96 (0.86–1) |
| Irregular Pleural Line | 22 (65) | 18 (36) | 0.65 (0.46–0.8) | 0.64 (0.49–0.77) | 26 (62) | 18 (36) | 0.62 (0.46–0.76) | 0.64 (0.49–0.77) |
| Pleural Effusion | 4 (12) | 0 (0) | 0.12 (0.03–0.27) | 1 (0.93–1) | 5 (12) | 0 (0) | 0.12 (0.04–0.26) | 1 (0.93–1) |
| Cavity | 3 (9) | 1 (2) | 0.09 (0.02–0.24) | 0.98 (0.89–1) | 5 (12) | 1 (2) | 0.12 (0.04–0.26) | 0.98 (0.89–1) |
| Pericardial Effusion | 6 (18) | 6 (12) | 0.18 (0.07–0.35) | 0.88 (0.76–0.95) | 8 (19) | 6 (12) | 0.19 (0.09–0.34) | 0.88 (0.76–0.95) |
| Hepatic Lesions | 1 (0.3) | 0 (0) | 0.03 (0–0.15) | 1 (0.93–1) | 1 (0.2) | 0 (0) | 0.02 (0–0.13) | 1 (0.93–1) |
| Periaortic Lymphadenopathy | 1 (0.3) | 0 (0) | 0.03 (0–0.15) | 1 (0.93–1) | 1 (0.2) | 0 (0) | 0.02 (0–0.13) | 1 (0.93–1) |
| Ascites | 2 (0.6) | 0 (0) | 0.06 (0.01–0.2) | 1 (0.93–1) | 2 (0.5) | 0 (0) | 0.05 (0.01–0.16) | 1 (0.93–1) |

* CI = confidence interval

† FASH = Focused Assessment with Sonography for HIV-Associated Tuberculosis

^ Refer to Table 1 for definitions of composite scores

## Discussion

In this proof-of-concept study designed to evaluate the performance characteristics of ultrasound to detect abnormalities associated with TB, we found that any thoracic or abdominal ultrasound abnormality had a high sensitivity (91%) but low specificity (46%) for bacteriologic TB. Thoracic ultrasound alone had slightly lower sensitivity (85%) and higher specificity (52%). Prespecified combinations of thoracic ultrasound findings increased specificity at the cost of sensitivity.

This study evaluated a well-characterized ambulatory population, including both patients with TB and people in whom TB had been excluded based on symptoms, chest radiograph, and sputum. Overall, our findings that thoracic ultrasound has high sensitivity and low specificity for TB are similar to results reported in prior studies [9, 11–14, 23]. Several previously published studies have reported on thoracic ultrasound in TB patients [11–14, 23], though all had small study populations, and most were in hospitalized patients. Only one other previous

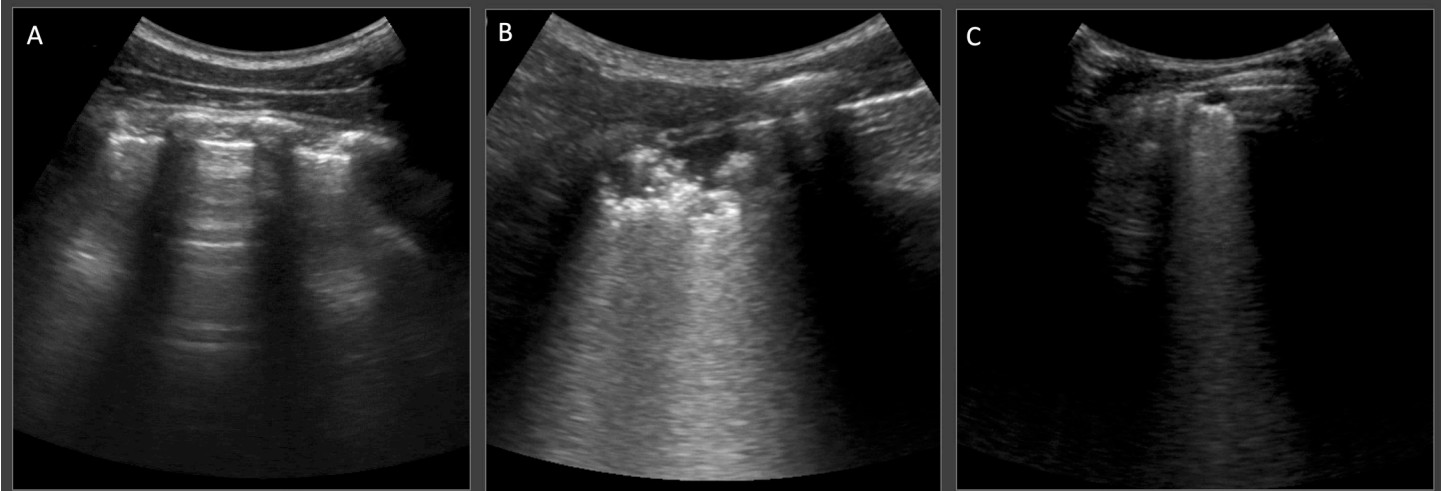

**Fig 1. Typical thoracic ultrasound findings.** Panel A demonstrates the appearance of normal air-filled lung, characterized by a bright white horizontal pleural line interrupted by rib shadows, and repeating horizontal reverberation artifacts known as A-lines. Panel B shows lung consolidation, characterized by subpleural echo-poor region greater than 10 mm in depth or length. Panel C demonstrates a small subpleural consolidation (SPC), characterized by a hypoechoic subpleural region less than 10 mm in depth and length, with distinct borders and a trailing comet-tail artifact.

study included participants without TB and reported on specificity [13]: this study of 102 patients (51 with PTB) in Italy reported ultrasound specificity of 92.2% for apical consolidations and 66.7% for small subpleural consolidations [13]. Most studies reported the presence of sonographic lesions in a high proportion of TB patients—for example, consolidation was detected in 78%–80% of TB patients [12, 13], SPC in 73%–80% [12, 13] and the presence of either consolidations or SPCs in 96%–98% [11, 12]. Cavity was detected infrequently in these studies, similar to our study. This represents an important limitation of ultrasound since cavity is a key radiological finding in pulmonary TB and increases the risk of transmission [24]. Overall performance did not appear to differ markedly between participants with and without HIV, although this subgroup analysis must be interpreted with caution given the small number of participants.

Few of our study participants had abnormal abdominal ultrasound findings, likely related to a relatively low burden of extrapulmonary TB in this ambulatory population that included HIV-negative people. Despite these low numbers and wide confidence intervals, the trend of high specificity observed for abdominal findings is similar to studies which have used ultrasound to assess for HIV-associated TB in the emergency department or inpatient setting [15–17]. In addition to its utility in the acute care setting, ultrasound has also been suggested to have a potential role in screening for extrapulmonary TB in those newly diagnosed with HIV through outpatient voluntary counseling and testing (VCT) [25].

The sensitivity of ultrasound for TB in our study is similar to the sensitivity of chest radiography for TB reported in the literature. For example, according to a WHO analysis, chest radiography with TB abnormalities has a pooled sensitivity of 85% and chest radiography with any abnormality has a pooled sensitivity of 94% [3]. However, specificity of ultrasound for TB in our study is lower than reported specificity of chest radiography. Pooled specificity of chest radiograph with TB abnormalities or any abnormality was 96% and 89%, respectively [3]. Some individual studies have reported lower specificity ranging from 63%–67% for any pathology and 67% for TB pathology [26, 27], which is similar to the specificity of predefined ultrasound composites in our study. The sensitivity of chest radiography in our study was lower than in published studies, which could be a function of our purposeful patient selection, our

use of a microbiologic reference standard, or the low number of cases with resulting wide confidence intervals. Computer-aided detection (CAD) of chest radiographs for TB, which is recommended by WHO and may be used for TB screening in community or primary health settings where specialized staff are not available, has variable performance characteristics, depending on the specific machine learning algorithm used and the positivity threshold applied. For example, a 2021 study of five CAD systems for TB reported that, at a fixed sensitivity of 90%, specificity ranged from 61%–74% [28]. A 2019 systematic review of CAD for TB found that, amongst the 13 clinical studies included, sensitivity ranged from 53%–100%, and specificity from 23%–98% [29].

The high sensitivity of ultrasound for TB demonstrated in this study suggests potential for its use as a TB triage tool. WHO recommends the minimum performance characteristics for a TB triage tool are sensitivity of 90% and specificity of 70% [30]. Our study was designed as a proof-of-concept study, and is too small to adequately investigate combinations of ultrasound findings which may improve specificity. However, future studies with larger samples could evaluate which combinations of ultrasound findings optimize specificity while maintaining acceptable sensitivity and develop a scoring system that incorporates combinations of findings. Use of point-of-care ultrasound could also be supported by artificial intelligence algorithms that guide the operator and identify abnormalities [31], allowing the technology to be used by providers with limited training. If developed at scale, this tool could contribute to improving TB diagnosis in settings where chest radiography is not available, or augmenting algorithms that include chest radiography.

The potential logistical advantages of ultrasound over chest radiograph for diagnosis of TB include its portability, safety profile, low cost, need for minimal consumables, and ability to function on battery power in remote settings [12]. CXR may not be available at lower levels of the health system in many high burden settings, and the portability and low cost of ultrasound could present a particular advantage in these settings. However, despite the potential advantages of ultrasound, there are a number of challenges to implementation of ultrasound as a diagnostic tool for TB disease, including the need for robust operator training, and the length of time required for the comprehensive scan technique used in this study [32].

Strengths of this study include use of a bacteriologic reference standard, recruitment of ambulatory participants in an HIV- and TB-prevalent region, and a robust ultrasound methodology that included expert review of all images. The primary limitation of this study is a small sample size. The study was originally designed to recruit 200 participants, but we were forced to suspend recruitment in March 2020 due to the COVID-19 pandemic. In addition, since the study was designed to focus on ambulatory patients, the results may not be generalizable to a hospitalized population. The lack of systematic testing for extrapulmonary TB could have resulted in misclassification. Finally, the ultrasounds were performed and interpreted under idealized rather than real-world conditions, which may have resulted in an overestimation of diagnostic accuracy, although this was in keeping with the proof-of-concept design.

## Conclusions

In this proof-of-concept study among ambulatory participants in a setting of high TB and HIV prevalence, a comprehensive ultrasound scanning protocol had high sensitivity but low specificity to identify bacteriologically-confirmed TB. Ultrasound has potential as a triage test for TB if specificity can be improved. Artificial intelligence systems could have a role if they can enable use by operators with limited training.

## Supporting information

**S1 Table. Sensitivity and specificity of ultrasound findings for bacteriologic TB in participants with and without HIV.**
(XLSX)

**S2 Table. Ultrasound findings in participants with bacteriologic TB compared to healthy participants.**
(DOCX)

## Acknowledgments

The authors wish to thank the individuals who consented to participate in this research; the team at Africa Health Research Institute, including Dickman Gareta, Njabulo Dayi, Njabulo Myeza, Mthokozisi Mnomiya, Thabani Mtshali, Zoey Mhlane, Farina Karim, and the Laboratory Team; and Tansy Edwards at the London School of Hygiene & Tropical Medicine. Butterfly Network provided an ultrasound device free of charge for use in the study but had no input into the design, conduct or reporting of the study.

## Author Contributions

**Conceptualization:** Matthew Fentress, Patricia C. Henwood, Priya Maharaj, Aaron S. Karat, Emily B. Wong, Alison D. Grant.

**Data curation:** Matthew Fentress, Patricia C. Henwood, Priya Maharaj, Philip Caligiuri, Aaron S. Karat, Emily B. Wong, Alison D. Grant.

**Formal analysis:** Stephen Olivier.

**Funding acquisition:** Alison D. Grant.

**Investigation:** Matthew Fentress, Patricia C. Henwood, Priya Maharaj, Mohammed Mitha, Dilshaad Khan, Nokwanda Ngcobo.

**Methodology:** Matthew Fentress, Patricia C. Henwood, Priya Maharaj, Aaron S. Karat, Emily B. Wong, Alison D. Grant.

**Project administration:** Anita Edwards, Dirhona Ramjit.

**Supervision:** Alison D. Grant.

**Writing – original draft:** Matthew Fentress, Alison D. Grant.

**Writing – review & editing:** Patricia C. Henwood, Priya Maharaj, Mohammed Mitha, Dilshaad Khan, Philip Caligiuri, Aaron S. Karat, Stephen Olivier, Anita Edwards, Dirhona Ramjit, Nokwanda Ngcobo, Emily B. Wong.

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
