## [Decision Letter · Decision Letter 0]

28 Mar 2022

PGPH-D-22-00300

High sensitivity of ultrasound for diagnosis of tuberculosis in adults in South Africa

Dear Dr. Fentress,

Thank you for submitting your manuscript to PLOS Global Public Health. After careful consideration, we feel that it has merit but does not fully meet PLOS Global Public Health’s publication criteria as it currently stands. Therefore, we invite you to submit a revised version of the manuscript that addresses the points raised during the review process.

We look forward to receiving your revised manuscript.

Kind regards,

Andrew D. Kerkhoff

Academic Editor

Journal Requirements:

1. Please amend your detailed Financial Disclosure statement. This is published with the article, therefore should be completed in full sentences and contain the exact wording you wish to be published.

ii). State the initials, alongside each funding source, of each author to receive each grant.

iii). State what role the funders took in the study. If the funders had no role in your study, please state: “The funders had no role in study design, data collection and analysis, decision to publish, or preparation of the manuscript.”

Additional Editor Comments (if provided):

Abstract

- Please add sub-headers to provide improve structure

- First sentence: please clarify if you mean all forms of TB (PTB, EPTB, or both) and among all persons, or restricted to HIV-positive persons only.

- Please add slightly more details about the reference standard. Was it sputum-based only? Were samples systematically collected? Please also briefly mentioned who performed the US exams (e.g., was this highly skilled/experienced radiologists?)

- I appreciate that numbers are small, but it would be appropriate add brief details to results as whether performance appeared to differ according to HIV status.

Results

- Please clarify how many participants were enrolled from the community survey compared to those from the TB facility. This has important implications on pre-test probability of TB disease and how advanced TB disease may be.

- I know the number of participants was small, but did you evaluate the overall sens/spec by HIV status? I think if even as a supplementary table, this is important data to report. However, I do appreciate that there is not sufficient statistical power to draw any firm conclusions from such sub-group analyses.

Discussion:

- Please add additional discussion regarding challenges of implementing US as a screening tool for TB disease. E.g., it took on average 41 minutes for highly skilled radiologists/pulmonologists. Further how available is US compared to CXR at lower health levels in high burden settings?

- Somewhere is the discussion it is important to note that accuracy may be overestimated compared to real-world settings, given 1) they were not read under real-world clinical demands/settings and 2) they were by highly experienced study staff and not clinicians or clinical officers at the study facilities.

- Under limitations, please also discuss the lack of systematic testing for EPTB to define the reference standard. This is especially important to mention given you are evaluating the diagnostic accuracy, at least in part, of abdominal ultrasound. This may have resulted in the misclassification of patients.

- You mention AI for US on two separate occasions in your discussion. On the first mention, are there any references that be provided that show initial promise for this approach?

- You also note use of US as a screening tool on multiple occasions. However, it appears the authors are referring to use as a triage tool (e.g., in persons who are actively seeking care and who are largely symptomatic). It’s a bit difficult to tease out as they note enrolment of participants they may represent a screening use case (survey participants) use and others that represent a triage use (health-facility participants). Please clarify.

Reviewers' comments:

Reviewer's Responses to Questions

**Comments to the Author**

1. Does this manuscript meet PLOS Global Public Health’s publication criteria? Is the manuscript technically sound, and do the data support the conclusions? The manuscript must describe methodologically and ethically rigorous research with conclusions that are appropriately drawn based on the data presented.

Reviewer #1: Yes

Reviewer #2: Yes

2. Has the statistical analysis been performed appropriately and rigorously?

Reviewer #1: Yes

Reviewer #2: Yes

3. Have the authors made all data underlying the findings in their manuscript fully available (please refer to the Data Availability Statement at the start of the manuscript PDF file)?

Reviewer #1: Yes

Reviewer #2: Yes

4. Is the manuscript presented in an intelligible fashion and written in standard English?

Reviewer #1: Yes

Reviewer #2: Yes

5. Review Comments to the Author

Reviewer #1: I read with interest this paper. Authors wrote an interesting and well presented paper. The role of ultrasound in Tb diagnosis is an important topic. Thanks for treat this topic. Below my suggestions

1. Introduction: updata data on Tb following tb report 2021 and introduce better why the ultrasound can be useful in tb diagnosis.

2. Methods and results: excellent, clear , statistical analysis is perfect. Well done

3. Discussion: add the potential role of ultrasound also in abdominal tb in HIV patients as showed in a paper from South Sudan, (see and cite Bobbio F, Di Gennaro F, Marotta C, Kok J, Akec G, Norbis L, Monno L, Saracino A, Mazzucco W, Lunardi M. Focused ultrasound to diagnose HIV-associated tuberculosis (FASH) in the extremely resource-limited setting of South Sudan: a cross-sectional study. BMJ Open. 2019 Apr 2;9(4):e027179. doi: 10.1136/bmjopen-2018-027179. ). Furthermore, give same global health proposal that came from your very very good paper

Reviewer #2: - A systematically selected convenience sample was used which are likely to skew the results towards a higher sensitivity. A group of patients already diagnosed with TB were part of the included group who will further skew the results as no negative TB cases are possible (hence limiting the specificity and the use as a screening tool). Please include how many of these patients were included in the final sample - if the majority of the TB positive patients are from this group, this study is unfortunately nothing more that a case series.

- Participants with suspected drug-resistant TB were excluded. Please include the definition of suspected drug-resistant TB and justify why they were excluded.

- Beside the small sample which resulted in an under powered study, there are two more limitations to the study that need to be acknowledged. Ultrasounds were performed by radiologists or pulmonologists which limits the generalizability of the study, specifically as TB are very prevalent in resource-limited settings where these expertise might not be available nor affordable. Secondly, the thoracic ultrasound took 32 minutes (and the abdomen a further 7 minutes), this is rather long to implement as a screening tool.

Minor comments:

- The term reference standard is preferred over gold standard.

- Please be consistent in the use of thoracic abnormalities versus pulmonary abnormalities.

- It is no surprise that the FASH scan performed poorly as half the sample were HIV-negative. The value of ultrasound as a screening test in ambulatory symptomatic people would be limited to pulmonary TB (i.e.thoracic ultrasound).

- The tables are rather long and the authors should consider to put ultrasound findings with limited numbers as supplementary material

6. PLOS authors have the option to publish the peer review history of their article (what does this mean?). If published, this will include your full peer review and any attached files.

**Do you want your identity to be public for this peer review?** For information about this choice, including consent withdrawal, please see our Privacy Policy.

Reviewer #1: No

Reviewer #2: **Yes: **Daniël J. van Hoving

---

## [Editor Report · Decision Letter 1]

30 May 2022

PGPH-D-22-00300R1

High sensitivity of ultrasound for diagnosis of tuberculosis in adults in South Africa

Dear Dr. Fentress,

Thank you for submitting your manuscript to PLOS Global Public Health. After careful consideration, we feel that it has merit but does not fully meet PLOS Global Public Health’s publication criteria as it currently stands. Therefore, we invite you to submit a revised version of the manuscript that addresses the points raised during the review process.

We look forward to receiving your revised manuscript.

Kind regards,

Andrew D. Kerkhoff

Academic Editor

Journal Requirements:

Additional Editor Comments (if provided):

I thank the authors for their resubmission and thoughtful revisions. I have reviewed the revised manuscript and I believe it has been substantially strengthened and can be accepted for publication pending addressing the minor points outlined below. Because there is no formal editing service during the publication process, I kindly ask the authors to carefully review their manuscript one final time for any grammar or typographical errors.

Title:

- Add “the” to title: High sensitivity of ultrasound for the diagnosis of tuberculosis in adults in South Africa.

- It would also be informative to make clear that this is a pilot/proof-of-concept study. Please consider adding a descriptive study title after the primary study title, (alternative wording also ok) e.g., "High sensitivity of ultrasound for the diagnosis of tuberculosis in adults in South Africa: a proof-of-concept study"

Abstract:

- Please break this sentence into two sentences or add semi colon: e.g., “We performed comprehensive thoracic and focused abdominal ultrasound examinations were performed by trained radiologists and pulmonologists on adults recruited from a community multimorbidity survey and a primary healthcare clinic in KwaZulu-Natal Province, South Africa. Sputum samples were systematically collected from all participants.”

- Please consider acknowledging the setting (e.g., South Africa) or that it is a high burden setting, e.g., “A comprehensive ultrasound scanning protocol in adults in a high TB burden setting had high sensitivity but low specificity to identify bacteriologically-confirmed tuberculosis.”

Author summary:

- First sentence: Please acknowledge that these limitations in diagnostic access are often at lower health care levels where patients may first present and clarify what is meant by diagnosis is meant by where TB diagnosis is routinely performed. Are they suggesting a high level of clinical diagnoses/empiric treatment due to lack of available diagnostic tools?

- For the following sentence, please slightly reframe language for improved clarity, as high sensitivity suggests a role as a triage tool to help rule-in TB. “The high sensitivity of ultrasound for tuberculosis in this study suggests a potential for ultrasound to be used for diagnosis of tuberculosis, particularly in areas where chest radiography is not accessible.”

Manuscript body:

- Please slightly revise the following sentence to remove the word major, “A 2021 systematic review of thoracic ultrasound for TB diagnosis found only six, mostly hospital-based studies that had methodological limitations, including small sample sizes and a lack of well characterized comparison groups.”

- Please slightly reframe this sentence to be consistent with a triage use case (rather than diagnosis), “If ultrasound has

appropriate performance characteristics for diagnosis of PTB, it could be incorporated into point-of-care algorithms to accelerate diagnosis and treatment of TB.”

- Please add a comma after which. “Survey participants who reported any TB symptom (cough, weight loss, night sweats, or fever) or had a CAD4TB score above 25 were asked to give a sputum specimen, which was divided into two in the laboratory”

-Table 1. Please reformat in a way that it is one single table (e.g., not two under a single Table 1 header)

- Please add a period to the end of the following sentence. “If radiological signs of previous TB were present, lower certainty was assigned to the classification of active TB”

- Under the “ultrasound findings” subsection, for consistency, I would report the findings pertaining to FASH/abdominal US in the same way you have for Thoracic US.

-Table 2. Please move the total/overall column to left side before the active TB participants column.

-Table 3. Is there any way to restructure this table so that there is clearer hierarchy/is easier to follow? For example could you group any thoracic pathology directly preceded or followed by the individual features and similarly for any FASH pathology? The table is difficult to follow as is. Additionally, I think it would also be helpful to either include definitions of different combination findings directly in the footnote or reference that they can be found in table 1.

-Table 4. Similar point to Table 3. Are there opportunities for restructuring the table in a way that may make nested findings clearer? Also, while I think this is useful/important data to include, I believe that this may be more appropriate as a Supplementary Table, especially as it may be slightly confusing to some reviewers as to how this differs from Table 3.

-The following sentence seems like it should be referenced. Are the relevant references the same as the following sentence? “Overall, our findings that thoracic ultrasound has high sensitivity and low specificity for TB are similar to results reported in prior studies.”

-For the following sentence, consider adding a very brief mention of how cavities are also associated with a high degree of infectiousness. “This represents an important limitation of ultrasound since cavity is a key radiological finding in pulmonary TB.”

- As this is the discussion section, please remove table reference from the following sentence, “Overall performance did not appear to differ markedly between participants with and without HIV (Supplementary Table 1), although this subgroup analysis must be interpreted with caution given the small number of participants.

- Please don’t capitalize Voluntary Counseling and Testing in the following sentence “…in screening for extrapulmonary TB in those newly diagnosed with HIV through outpatient Voluntary Counseling and Testing (VCT).”
---

## [Editor Report · Decision Letter 2]

12 Sep 2022

High sensitivity of ultrasound for the diagnosis of tuberculosis in adults in South Africa: a proof-of-concept study

PGPH-D-22-00300R2

Dear Dr. Fentress,

We are pleased to inform you that your manuscript 'High sensitivity of ultrasound for the diagnosis of tuberculosis in adults in South Africa: a proof-of-concept study' has been provisionally accepted for publication in PLOS Global Public Health.

Best regards,

Andrew D. Kerkhoff

Academic Editor